# Fibroblast Growth Factor 21 as a Potential Biomarker for Improved Locomotion and Olfaction Detection Ability after Weight Reduction in Obese Mice

**DOI:** 10.3390/nu13092916

**Published:** 2021-08-24

**Authors:** Nicole Power Guerra, Alisha Parveen, Daniel Bühler, David Leon Brauer, Luisa Müller, Kristin Pilz, Martin Witt, Änne Glass, Rika Bajorat, Deborah Janowitz, Olaf Wolkenhauer, Brigitte Vollmar, Angela Kuhla

**Affiliations:** 1Rudolf-Zenker-Institute for Experimental Surgery, Rostock University Medical Centre, Schillingallee 69a, 18057 Rostock, Germany; nicole.guerra@uni-rostock.de (N.P.G.); alisha.parveen@med.uni-rostock.de (A.P.); daniel.jenderny@uni-rostock.de (D.B.); luisa.mueller2@uni-rostock.de (L.M.); brigitte.vollmar@med.uni-rostock.de (B.V.); 2Department of Anatomy, Rostock University Medical Centre, Gertrudenstraße 9, 18057 Rostock, Germany; martin.witt@med.uni-rostock.de; 3Department of Systems Biology and Bioinformatics, University of Rostock, Ulmenstraße 69, 18057 Rostock, Germany; david.brauer@uni-rostock.de (D.L.B.); olaf.wolkenhauer@uni-rostock.de (O.W.); 4Department of Psychosomatic Medicine and Psychotherapy, Rostock University Medical Centre, Gehlsheimerstraße 20, 18147 Rostock, Germany; 5Centre for Transdisciplinary Neurosciences Rostock (CTNR), Rostock University Medical Centre, Gehlsheimerstraße 20, 18147 Rostock, Germany; 6Department of Psychiatry, University of Greifswald, Ellernholzstraße 1-2, 17489 Greifswald, Germany; kristin.pilz@med.uni-greifswald.de (K.P.); deborah.janowitz@helios-gesundheit.de (D.J.); 7Institute for Biostatistics and Informatics, Rostock University Medical Centre, Ernst-Heydemann-Straße 8, 18057 Rostock, Germany; aenne.glass@med.uni-rostock.de; 8Department of Anesthesiology and Intensive Care Medicine, Rostock University Medical Centre, Schillingallee 35, 18057 Rostock, Germany; rika.bajorat@med.uni-rostock.de; 9Leibniz-Institute for Food Systems Biology, Technical University of Munich, Lise-Meitner-Straße 34, 85354 Freising, Germany

**Keywords:** FGF21, treadmill, time restricted feeding, machine learning, behaviour, diet-induced obesity, feature selection, high-fat diet

## Abstract

Obesity is one of the most challenging diseases of the 21st century and is accompanied by behavioural disorders. Exercise, dietary adjustments, or time-restricted feeding are the only successful long-term treatments to date. Fibroblast growth factor 21 (FGF21) plays a key role in dietary regulation, but FGF21 resistance is prevalent in obesity. The aim of this study was to investigate in obese mice whether weight reduction leads to improved behaviour and whether these behavioural changes are associated with decreased plasma FGF21 levels. After establishing a model for diet-induced obesity, mice were subjected to three different interventions for weight reduction, namely dietary change, treadmill exercise, or time-restricted feeding. In this study, we demonstrated that only the combination of dietary change and treadmill exercise affected all parameters leading to a reduction in weight, fat, and FGF21, as well as less anxious behaviour, higher overall activity, and improved olfactory detection abilities. To investigate the interrelationship between FGF21 and behavioural parameters, feature selection algorithms were applied designating FGF21 and body weight as one of five highly weighted features. In conclusion, we concluded from the complementary methods that FGF21 can be considered as a potential biomarker for improved behaviour in obese mice after weight reduction.

## 1. Introduction

Obesity is reaching a global epidemic scale and is defined as abnormal or excessive body fat accumulation [1]. Already in 1989, Kaplan described the “Deadly Quartet” of abdominal obesity, hypertension, hyperglycemia, and hypertriglyceridemia with accompanying low concentrations high-density lipoprotein cholesterol [2]. A promising candidate for reducing plasma concentrations of cholesterol and triglycerides is Fibroblast Growth Factor (FGF) 21 [3,4]. The hormone FGF21 is associated with fatty acid oxidation, lipolysis, increased energy dissipation, and hence weight reduction [5,6,7]. Astonishingly, obese humans and mice exhibit exceedingly high levels of circulating FGF21 plasma concentrations when compared to lean patients or wild type mice [8]. This evidence sparks the idea of whether FGF21 can be considered as a biomarker in obesity [9,10].

In obesity, food regulation and energy expenditure are heavily disturbed [1] (Figure 1). Obesity-related low-grade inflammation in adipose tissue is assumed at the origin of the disease, later leading to a neuroinflammation as described in Figure 1 [11,12]. The resulting gliosis is hypothesised to dysregulate endocrine balance in the hypothalamus, and thereby the in hypothalamus–pituitary axis (HPA), leading to altered nutrition intake [12,13]. Since FGF21 is intricately connected to nutritional regulation [14], the inflammation is thought to reduce FGF21 sensitivity [15]. As one consequence, body weight and fat increases, which in turn boosts low-grade inflammation. The inflammatory presence effectively leads to higher FGF21 production and a further imbalance of nutritional regulation, thus closing the vicious circle of obesity. Therefore, the increase in weight and in FGF21 concentration is described as an FGF21-resistance state [15,16].

In addition, the presence of obesity reveals further impact on cognitional animal behaviour depending on the diet and nutrition model [17]. For example, in an animal model of diet-induced obesity (DIO) and in mice receiving a high-fat diet (HFD), olfactory dysfunctions and anxiety-like behaviour are shown to be favoured, which may lead to reduced activity per se [18,19]. In this context, it is described that obesity in adolescents aggravates physical inactivity and vice versa, which consequently increases the risk of overall and abdominal obesity in adulthood [20]. Thus, to overcome the vicious circle of obesity, intervention approaches such as treadmill exercise, a change in diet or fasting are suitable and common methods to lower FGF21 concentration while increasing FGF21 sensitivity [21,22,23]. The purpose of the study was to investigate in obese mice whether weight reduction leads to improved behaviour and whether these behavioural changes are associated with altered plasma FGF21 concentrations. We aimed to investigate if FGF21 may be considered in this context as a biomarker for behavioural improvement after weight reduction. To investigate and evaluate this hypothesis, different analysis tools are on hand. Besides statistical analysis, machine learning (ML) models are able to improve prediction accuracy by discovering relevant features of high complexity [24]. In this study, we determine the weighted features by using three different feature selection (FS) algorithms which eliminate irrelevant or redundant features from the original data set [25]. Accordingly, informative features remain which in turn might indicate their biological significance. However, regarding smaller data sets with mouse studies, the repetition of experiments is low and group size is limited. Therefore, applied models are often prone to biasing issues due to the small sample size [26]. To target the problem at hand, we applied multiple classification models to ensure validation by quantity. The novelty of this study is the combination of the ML method with FS, considered as an additional tool, and statistical methods on a small data set of behavioural parameters to determine whether FGF21 may be a biomarker for weight loss in obese mice.

## 2. Materials and Methods

### 2.1. Experimental Design

For the experiments, 90 female C57BL/6J mice aged 4 weeks were purchased from Charles River (Sulzfeld, Germany). Mice were kept in standard cages with 5 animals per cage, in a temperature-controlled room (21 ± 3 °C) with a 12/12 h day-night cycle (lights on from 06:00 a.m. to 06:00 p.m.). Randomisation was not performed at this step as mice were purchased and equally handled. To establish the model of DIO, all 90 mice initially received a high-fat diet (HFD; D12492; Research Diets, Lane, USA) for 6 months. For the intervention in the following 6 months, cages were arbitrarily divided into six groups. The first group (*n* = 15) remained on HFD, hereinafter referred to as “HFD/HFD”. The second group additionally participated in TM exercise (TSE System, Treadmill 303401; *n* = 15), referred to as “HFD/HFD + TM”. The third group was also trained on treadmills and additionally received a time restriction on food (TRF) intake after the first three months of the intervention (*n* = 15), designated as the group “HFD/HFD + TM + TRF”. In the fourth group, HFD was changed to a low-fat diet (LFD; D12450J; Research Diets, Lane, USA) (*n* = 15) and is hereafter referred to as “HFD/LFD”. The fifth group also switched to LFD accompanied by TM exercise (*n* = 15) and named “HFD/LFD + TM”. The last group (*n* = 15) underwent all three interventions and is designated as the group “HFD/LFD + TM + TRF”. Graphical illustration of the experimental design is shown in Figure 2A. During the experiments, each mouse had ad libitum access to fresh water.

### 2.2. Intervention Parameters

After the establishment of a DIO model in mice, different intervention strategies were performed in order to evaluate their effectiveness.

#### 2.2.1. Diet Change to LFD

Mirroring a healthier food intake, 45 mice changed as a first intervention parameter to an LFD containing 10% fat, 20% protein, and 70% carbohydrates and matching the HFD in structure of lards and protein composition. In contrast, HFD is composed of 60% fat, 20% protein, and 20% carbohydrates [27].

#### 2.2.2. TM Exercise

After the dietary change was accomplished, the second intervention parameter, TM exercise, was implemented for *n* = 60 mice. The protocol was adapted after Marinho et al. (2018) with modifications to mimic human patterns [28]. TM was performed twice a week in groups of five (Figure 2B). The velocity of the treadmill was set according to the speed of the mouse that performed as the slowest.

The treadmill program was composed of seven stages beginning with a five-day training of 10 min/d with a speed of 0.1 m/s and 0° incline in order to adapt to the treadmill. Favouring optimal lipid oxidation rather than carbohydrate combustion, the maximal lactate steady state for each mouse had to be determined [29]. Hereby, mice were subjected in the second stage to an interval training starting with 0.1 m/s and 0° incline. Increments of 0.05 m/s were adjusted every 300 s until voluntary exhaustion of mice. The maximum achieved velocity is defined as the workload, whereas 60% of the workload specified the speed for the endurance training. The incremental load test was performed once prior to endurance training [30]. The third stage provided an eight-week endurance training with two exercise days per week. At the beginning, mice started with 15 min/d and 0° incline endurance training at the previously obtained velocity. The duration increased by 15 min every 2 weeks up to a maximum of 60 min endurance training twice a week. Then, a second incremental load test (fourth stage) was performed to adjust the maximum workload and as a result the endurance speed. With the adjusted velocity a second eight-week endurance training was accomplished (fifth stage) with 60 min exercise twice a week. As the sixth stage, a third incremental load test was conducted followed by an adapted third endurance training. Treadmill exercise was maintained until the sacrifice of mice.

On days with behavioural experiments, no endurance training was performed to avoid any interference for upcoming analyses. Animals were excluded from TM exercise if they either did not accept the acclimatisation phase in the first stage or were no longer willing to perform certain exercises.

#### 2.2.3. TRF

Temporary restriction of food is described as beneficial against obesity and metabolic disorders [21,31]. Therefore, a third intervention parameter, TRF, was applied for *n* = 30 mice after the implementation of the third phase of TM exercise (Figure 2C). The protocol was adapted after Hatori et al. (2012) [21] and maintained for 3 months until the sacrifice of mice. Food access was regulated by using an autofeeder (EHEIM, Deizisau, Germany) whose opening was enlarged. Food drop was set at 11 p.m. and controlled via a webcam with infrared light. At 7 a.m., mice were transferred to fresh cages with water supply, filled autofeeder, and no enrichments. To equalise animal handling to all groups, *ad libitum* fed mice were also transferred daily.

### 2.3. Behaviour Experiments

#### 2.3.1. Buried Pellet Test (BPT) and Surface Pellet Test (SPT)

As cellular dynamics are modified in the olfactory bulb due to obesity, scent abilities might also be influenced [32]. Therefore, buried and surface pellet tests were conducted after Dragotto et al. (2019) [33] and Lehmkuhl et al. (2014) [34]. Briefly, mice were acclimated to a piece of sweetened pellet (Honey Llama Loops, Kellogg Company, EU) two days prior to testing. After overnight fasting of 9 h to a maximum of 16 h in their home cages, a single mouse was transferred to an ethanol wiped cage with 1 cm embedding. Each mouse was habituated for 1 h alone in a separated room. For the test, a new ethanol wiped cage was filled with 3 cm embedding and the cereal was buried 0.5 cm below the bedding surface next to one corner of the cage. Then, the subject was placed in the cage and latency time to uncover and lick or eat the pellet was measured. If the mouse did not find the pellet within 300 s, a score of 300 s was recorded. The test procedure was repeated on the subsequent day except that the cereal was placed on the surface (SPT). Thereby, motor deterioration and visual clues for finding the pellet were excluded [35].

#### 2.3.2. Elevated plus Maze (EPM)

The EPM is a widely used maze providing information about anxiety-related behaviour in rodents as mice have a natural aversion to open areas [36,37]. After recovery of at least 24 h from the SPT, the EPM protocol was performed after Komada et al. (2008) [38]. The 60-cm-high grey EPM consists of two open arms (6 cm in width, 40 cm in length), and perpendicular to the open arms are two closed arms of the same dimensions with walls of 14.5 cm high. The cross at the centre of the four arms consists of a 6 cm × 6 cm square, where a camera system is positioned 1 m above the maze (Camera CCA1300–60 mg, Basler, and lens 15E, Computar, Japan). Prior to testing, animals were kept at least 1 h in the behaviour room. Then, the subject was placed in the ethanol wiped maze and was recorded for 300 s. Hereafter, the mouse was placed back in the home cage and the maze was cleaned with ethanol for the subsequent animal. All sessions were measured by using EthoVision XT 11.5 software (Nodulus Information Technology). The following parameters were evaluated: cumulative duration in open and closed arms (%), cumulative duration in centre (%), arm entries (%), centre entries (%), total distance in maze (cm), mean velocity (cm/s), and vertical activity by counting and adding manually all leanings, rearing, and jumps. Graphical visualisation was displayed over 300 s by using the EthoVision XT 11.5 software. Red colour reveals highest residence time and blue colour indicates lowest duration. Subsequently, movement patterns of obtained images were manually compared. Visited arm entries were counted and were assigned as follows: centre only, centre + one closed arm entry, centre + two closed arm entries, centre + two closed and one open arm entries, all areas.

#### 2.3.3. Open Field (OF)

In the OF, not only can anxiety-like behaviour be observed, but also locomotor and exploration activity [39,40]. The protocol was performed after Seibenhener et al. (2015) [41]. The OF consists of a 50 cm x 50 cm square with 40 cm high walls whereby the square was divided virtually in the software into 16 small fields. The inner four squares represented the open area and were defined as centre. The remaining fields were specified as outer areas. In accordance with the EPM protocol, equal procedures were performed. The following parameters were evaluated: cumulative duration in centre and outer area (%), field entries (%), total distance in maze (cm), mean velocity (cm/s), and vertical activity by counting manually all rearing and jumps. Between both experiments, mice had a rest of a minimum of 3 h. Heatmap was created in accordance with EPM protocol. Visited area was examined and classified according to the following: only corners, outer area with less crossings, outer area with more crossings, outer area with half of centre visited, all areas visited.

### 2.4. Weight Control and Euthanasia

Body weight was measured weekly (Kern PCB, Lübeck, Germany) and final body weight was monitored prior to euthanasia. Under anaesthesia (5 vol.% isoflurane; Baxter, Unterschleißheim, Germany) the mice were exsanguinated retrobulbarly and thereby, blood was collected. Thereafter, a laparotomy was performed. The heart was punctured and perfused with 0.9% NaCl (Serag-Wiessner, Naila, Germany) with a flow rate of 2.59 mL/min for 12 min. The visceral and subcutaneous flanked fat deposits were harvested and weighed, and blood plasma was collected.

### 2.5. FGF21 ELISA of Blood Samples

FGF21 ELISA was performed following manufacturer’s description (ab212160, abcam, Berlin, Germany). All plasma samples were diluted 1:5.

### 2.6. Statistics: Multiple Comparisons of Means

Statistical analysis was performed with GraphPad Prism 8.0.1 (GraphPad Software Inc., San Diego, CA, USA). The data were first checked for normality and lognormality with a Shapiro–Wilk test. In the case of ‘vertical activity’ in EPM and OF, the data was tested with Kolmogorov–Smirnov test, as observations were manually counted. For lognormal distributed data, the data set was transformed according to the formula *Y = log(Y)*. The ROUT method based on false discovery rate (*Q* = 0.01) was used to identify and remove outliers if possible and necessary.

If the data were normally distributed, One-Way ANOVA was performed. Homogeneity of group variances was checked with Bartlett’s test. For homogenous data, an ordinary one-way ANOVA was performed followed by Tukey’s post hoc test for multiple comparisons of means. Otherwise, Brown–Forsythe and Welch ANOVA were performed followed by Tamhane’s T2 post hoc test for multiple comparisons of means. If the data was not normally distributed, Kruskal-Wallis test with Dunn’s post hoc test for multiple comparisons was performed. For the ML models classification report, significance of differences was tested by Wilcoxon Signed Rank Test. Data are presented as mean ± standard deviation (SD) and statistical significance was set at *p* < 0.05. For further details, please see figure legends.

### 2.7. Data Analysis

#### 2.7.1. Dimensionality Reduction

Analysis of FGF21 concentration, body composition, BPT, EPM, and OF yielded 32 observations with *n* = 83 mice divided into six groups (Figure 3, reduction of *n* = 83 was due to the exclusion criteria and death dropout of mice). Missing values in the data set were filled by calculating a stratified average value depending on *y*, where *y* represents the six intervention groups. For each task, data was correspondingly preprocessed and afterwards split into *y* (intervention groups) as a dependent variable and *x* (all other data; Appendix A). To correlate all 32 observations, Pearson’s correlation was performed where linear relationship between two variables is measured. In the heatmap, plotted red colour (1.00–0.70) indicates a strong positive correlation, blue colour (−0.70–−1.00) reveals a strong negative correlation, whereas light colour above 0.40 or under −0.40 indicates moderate correlation [42]. To reveal the relation between the six intervention groups, Principal Component Analysis (PCA) was conducted. To perform PCA on a dataset where observed entities *n* were smaller than the observations on variables *p* (*n < p*), we used a modified PCA implementation in Python with svd_solver = ‘arpack’ [43].

#### 2.7.2. Machine Learning Approach

To predict whether FGF21 is a putative biomarker for improved behaviour after weight reduction, the following procedure was implemented: To address the low sample size, three different FS algorithms were applied for the selection of the putative features, namely Chi-Square (Chi2), Ridge Regularisation (RIDGE), and Recursive Feature Elimination (RFE) for the selection of the putative features [44]. The selected features were visualised in a Venn diagram with InteractiVenn provided by Heberle et al. (2015) [45]. To assess the viability of the FS algorithms, eight different supervised ML models were constructed on both the FS data set (only the selected features) and the original, non-FS data set. To train the models, both data sets were split into training (80%) and test (20%) data sets. Then, eight different supervised ML algorithms were used (Logistic Regression, Support Vector Classifier (SVC), Decision Tree, Naive Bayes, Random Forest, Gradient Boosting (Gradient B.), Stochastic Gradient Descent (SGD), and Neural Network). For further evaluation, each model was verified using 6-fold cross-validation (CV). We opted against 10-fold CV since, on the one hand, we are working with a small data set and, on the other hand, we are considering six different experimental groups as the study design. Ultimately, accuracies (=recall), precision, and F1-scores were compared between the non-FS data set and the FS data set. A high F1-score indicates that a model exhibits low false positives and low false negatives. The full classification report with weighted averages for each model is displayed in Appendix A.

#### 2.7.3. Implementation in Python

For the analysis, Python (version 3.8) was used. The full data table and all coding sections were upload on 21 April 2021 and can be accessed under https://github.com/IEC-2020/Intervention, (accessed on 21 April 2021). All methods, libraries and classes used to accomplish this work are summarized in Table 1. Descriptions of all observations are listed in Appendix A.

## 3. Results

### 3.1. Effect of LFD and TM Exercise on Body Composition and FGF21

A continuous administration of HFD led to a high increase in body weight within the first 6 months (Figure 4A). After the introduction of intervention parameters such as a diet change to LFD, TM exercise, and TRF, only the dietary adjustment led to weight loss within a few weeks (Figure 3A). Ultimately, body weight and fat weight are about 50% lower when comparing HFD/HFD groups to HFD/LFD groups Figure 4B–D; B: *p* ≤ 0.0012 for HFD/HFD groups vs. HFD/LFD groups; C, D: *p* < 0.0001 for HFD/HFD groups vs. HFD/LFD groups, respectively). The same groups are also prominent regarding FGF21 concentration, exhibiting a significant reduction in the HFD/LFD groups (Figure 4E). HFD/LFD + TM revealed the lowest FGF21 plasma concentration with 366.8 ± 281.7 [pg/mL] (Figure 3E; *p* < 0.0001 for HFD/LFD + TM vs. HFD/HFD). Noteworthy, the HFD/HFD + TM + TRF group also displayed a significant reduction in FGF21 concentration compared to the HFD/HFD + TM group (Figure 4E; *p* = 0.0053). This phenomenon, where the combination of HFD with TM and TRF led to a significant change, was not observed in the other surveyed parameters. In conclusion, the transition to LFD exhibited the most significant effects, and the HFD/LFD + TM group was emphasised through the lowest FGF21 plasma levels.

### 3.2. The Combination of LFD and TM Exercise Improves Behavioural Parameters

The combination of diet change to LFD and TM exercise also exhibited the highest effect in all three behavioural experiments, namely, EPM, OF, and BPT. In the EPM, activity was measured by recording the total distance travelled in the maze, the vertical activity of mice, the mean velocity while exploring the maze, the immobility of mice, the presence of mice in the maze and entries of every arm (Figure 5A–G). The HFD/HFD group showed reduced overall activity with the lowest mobility pattern and most time spent in closed arms (Figure 5A,E,F). Contrarily, HFD/LFD + TM reveals the highest activity with less immobility time and more presence in open arms (Figure 5A,E,F; immobility (E): *p* = 0.0010 vs. HFD/HFD, cumulative duration (F): *p* ≤ 0.0039 vs. HFD/HFD). These findings suggest less anxiety-related behaviour in the HFD/LFD + TM group with overall increased activity. In the OF, the same behavioural parameters were exhibited as in EPM, whereas the closed arm is represented as the outer area and the open arm as the centre (Figure 6A–G). HFD/LFD + TM revealed the highest mean velocity and the lowest immobility pattern when compared to all other groups underpinning an increased activity pattern (Figure 6D,E; mean velocity (D): *p* ≤ 0.0022 vs. all groups; immobility (E): *p* ≤ 0.0045 vs. all groups). Correspondingly, in the BPT the same group became prominent (Figure 7A). In three groups, namely HFD/LFD, HFD/LFD + TM and HFD/HFD + TM + TRF, 75% of the mice finished the experiment within 67 s. Interestingly, the parameter “food restriction” led to a significant improvement of smell abilities in the HFD/HFD + TM + TRF group (*p* = 0.0039 vs. HFD/HFD + TM) but not in combination with LFD. Notably, every group had mice that did not find the pellet in the required time. In addition, the HFD/HFD + TM group showed difficulties in finding the pellet resulting in the highest latency times (Figure 6B).

### 3.3. HFD/LFD and HFD/LFD + TM Are the Most Prominent Intervention Groups

To further assess the relation between the obtained data, two different statistical methods were applied. As a first approach, Pearson’s correlation was applied pairwise between all 32 observations from all experiments and is represented as a heatmap (Figure 8). Several strong positive (0.7–1) and negative correlations (−1–−0.7) are observed among certain parameters, such as body weight (1) to fat weight (rows 2,3), and finding the pellet in the BPT (row 5) to latency to lick or eat the pellet in the BPT (row 6), and vice versa. In a biological context, these correlations are causal; since body weight is manipulated by an HFD, fat deposits will also be affected. There is also causality between finding the pellet in the BPT and decreased time to eat or lick the pellet. Interestingly, strong negative correlations are revealed between FGF21 concentration (0) vs. mean velocity (row 28, OF) or distance moved in the OF (row 29). The observation implies if FGF21 concentration is increased, the velocity and distance travelled are minimised and vice versa. This plot demonstrates a variety of strong positive and negative correlations in a reasonably clear diagram highlighting parameters such as FGF21 concentration, body weight, mean velocity, and distance moved in mazes.

As a second approach, PCA was performed to find potential clustering between the intervention groups and to reveal new information about similarity (Figure 9). The intervention groups were used to colour the dot plot. Since Principal Component 1 (PC1) accounts for 74.68% of variances, the distance between data points on the *x*-axis represents a larger difference than on the PC2 axis, which accounts for 10.40% of variances. PC scores showing vertical “clusters” exhibit less variance and thus more similarities. The most striking PC scores are from the groups of HFD/LFD + TM (yellow), and partially of HFD/LFD (green) which are located around −0.2 of PC1. The dispersion of the HFD/LFD + TM group is the densest and exhibits a vertical “cluster” indicated by a dotted circle. These findings suggest that the parameters of dietary change to LFD in combination with TM display the greatest similarity to all intervention groups. The importance of the HFD/LFD + TM group is underpinned by behavioural analysis revealing improved performance. Additionally, HFD/LFD and HFD/LFD + TM are assumed to be closer related to each other when compared to the other groups, resulting in a greater impact of these interventions on the data set.

### 3.4. FGF21, Body Weight, Olfactory Detection, and Mobility Pattern Are Highly Weighted Features

To corroborate the role of FGF21 as a putative biomarker for improved behaviour after weight reduction, a supervised ML approach was used. Three differently operating FS algorithms were applied (Chi2, RFE, and RIDGE) to select key features and to avoid overestimation [44], see Appendix A for all selected features. Afterwards, the selected candidate features were visualised in a Venn diagram yielding eight common features, namely FGF21 concentration, body weight, subcutaneous flanked fat, visceral fat, latency to eat or lick the pellet in the BPT and SPT, vertical activity, and mobility time in the OF (Figure 10A). Since fat deposits correlate strongly with body weight, as shown in the Pearson’s correlation, these three features can be reduced to the feature ‘body weight’, providing a non-invasive parameter to measure. To further restrict the size of the common features, the crucial parameter of the olfactory analysis is the latency time in the BPT, since the SPT does not give any information about olfactory performance, but only about possible motor impairments. Therefore, the final five common features are FGF21 concentration, body weight, latency to eat or lick the pellet in the BPT, vertical activity, and mobility time in the OF.

To validate the performance of the FS algorithms, a classification report with accuracy, mean accuracy in 6-fold CV, precision, and F1-scores was calculated for each ML model on both data sets (Figure 10B). ML models based on the eight selected features revealed significantly higher accuracies, CV, precision and F1-scores with *p* = 0.0078 compared to the non-FS, highlighting Neural Network above all (Figure 10C). On the one hand, a lower accuracy implies that most of the ML algorithms are not resulting in a reasonable model. On the other hand, a higher model accuracy indicates that either the feature selected data set is more suitable for the models, or that an overfitting phenomenon emerges. However, it is more likely that the eight common features are the main parameters in the data set and therefore highly weighted. Collectively, these observations support that FGF21, among the other selected features, is a putative biomarker for improved behaviour after weight reduction in the dataset.

## 4. Discussion

Exercise, dietary adjustments, or time-restricted eating are, to date, the only successful long-term treatments against obesity in humans and mice. The aim is to restore the balance between disturbed energy dissipation and energy intake [20,46]. In this process of energy homeostasis, FGF21 is involved by modulating the metabolism in healthy individuals as well as in obesity and could function as a putative biomarker for improved behaviour after weight reduction in obese mice [10,47]. Additionally, obese individuals displayed altered behaviour regarding physical activity, olfaction, and anxiety [19,20]. The purpose of the study was to assess whether FGF21 is still valid as a biomarker after weight reduction in HFD-mice using behavioural parameters.

We revealed that dietary change to LFD was able to counteract obesity in terms of body weight and fat reduction which is in line with previous studies [48]. Although endurance TM training in multiple studies has been shown to lead to weight reduction [23,49], other studies found no weight reduction [50]. The varying findings may probably result from different treadmill intensities and durations protocols [51]. Thus, in our study, a dietary change to LFD showed a significant impact on weight reduction compared to a moderate TM approach.

The intervention parameter TRF is also reported to reduce body weight and fat mass by coupling food intake to circadian rhythm. Therefore, food consumption is restricted to the active nocturnal phase of the mice and reveals a lasting effect on the metabolic status of the liver [21,52]. However, the study design of our intervention did not incorporate a group which only combined TRF with an HFD or LFD. As a result, we were unable to observe positive effects of TRF in terms of weight reduction. Potentially, the weight reduction in the HFD group with TM and TRF might have been counterbalanced by the muscle gain, as aerobic training led to up to 6% hypertrophy of the quadriceps in humans [53]. Consequently, the weight reduction effects of the additional intervention of TRF were absent.

Nevertheless, TRF exhibited positive effects in combination with HFD and TM regarding FGF21 concentration and weight reduction [54]. One consequence of obesity is FGF21 resistance, which is described by high circulating FGF21 concentrations and greatly increased body weight [16]. We assume, since FGF21 has a circadian rhythm which is disrupted by an HFD, TRF rebalances the oscillation of FGF21 by coupling the food intake in a daytime-dependent manner [21,55]. This potential restoration of rebalance by the TRF regimen was shown by the recovery of an HFD-induced dysregulation of the oestrous cyclicity and FGF21 signalling has been proposed as a key player [56]. Therefore, we speculate that the decrease of FGF21 in the HFD group including TM and TRF might be a hint for the beneficial effect of TRF, as employment of TM alone did not lead to a decrease of circulating FGF21 concentrations. However, physical exercise was shown to reduce FGF21 concentration and was proposed to recover FGF21 sensitivity in obese mice and to rebalance the metabolic interaction between adipose tissue, liver and skeletal muscle [23]. This favourable impact on reducing FGF21 concentration by TM is not observed in our data with TM alone but in combination with a dietary change to LFD.

Furthermore, dietary change to LFD with TM resulted in less anxiety-related behaviour, overall higher activity, and better olfactory abilities when compared to HFD mice. Exercise training was reported to reduce anxiety sensitivity by modulating stressors to the HPA axis [57,58]. Among the stressors is the HFD, which leads to increased FGF21 concentrations as a nutritional response. FGF21 is in turn involved in metabolic stress processes and has been also described as a stressor [14]. It has been proposed that FGF21 could directly influence the hypothalamus and thus stimulate the HPA axis [59]. We suspect that, triggered by an HFD, a stress-induced increase in FGF21 concentration negatively modulates the HPA axis. This modulation is rescued by TM, which is manifested by a reduction in anxiety-related behaviour. Apparently, the positive effect on behaviour is not solely attributable to TM, otherwise, we would have observed this beneficial effect in combination with HFD and TM. Rather, the combination of LFD with TM is decisive; in other words, the combination of a reduction in nutritional stress—which is accompanied by a reduction in inflammation—and the positive effect resulting from physical activity are the pivotal factors for a promising intervention against behavioural dysfunction in obesity.

Concerning behavioural analysis, we assumed that TRF alone was beneficial [60], but we showed that the combination of dietary change to LFD and TM was also conducive. We also expected at least a similar effect when combining all three intervention strategies, if not even a magnified result. Surprisingly, the positive effect in behavioural analysis achieved by LFD and TM was diminished in combination with TRF, especially in the OF. Our group showed in a previous study that lifelong caloric restriction—a model for combatting obesity in terms of calorie reduction—led to more anxiety-related behaviour in EPM but a significant increase of working memory in female mice [61,62]. Gathering the information, we suspect that TRF is likely to promote anxiety-related behaviour and reversing the beneficial effects of TM in combination with LFD. Nevertheless, TRF introduces a significant modification in olfactory detection ability emphasised in the HFD group with TM and TRF. As HFD causes deterioration of odour recognition, odour discrimination and odour-dependent learning, TRF restores olfactory odour recognition [63]. In this context, an increase of olfactory sensitivity was observed in rats by an intracerebroventricular injection with orexin, an anorexigenic molecule imitating a fasting state [64]. The group with a dietary change to LFD and TM revealed similar olfaction improvements as in the HFD group with TM and TRF, thus also supporting the conclusion of a recovery of olfactory deficits caused by an HFD. Collectively, the increased activity pattern in the OF, the decreased anxiety-related behaviour in the EPM, and the restoration of olfactory recognition suggest that the combination of diet change to LFD and TM represents the most effective intervention against behavioural dysfunctions in obesity.

To further investigate the relationship between the interventions groups, PCA was used as an exploratory tool [43]. The observations of PCA indicated that the group with diet change to LFD with TM showed the fewest internal variance, underscoring the importance of this group. Given that the group which performed only a change to LFD partially forms a “cluster” with the previously described intervention group, there is an indication that dietary change can be considered to be more influential compared to the TRF intervention parameter against obesity. Furthermore, we used PCA as an additional tool to support previous findings, and explicitly not as a stand-alone approach.

Using machine learning methods, we aimed to strengthen the hypothesis of whether FGF21 persists as a putative biomarker for improved behaviour after weight reduction. Indeed, we identified FGF21, body weight, odour detection and the activity pattern in the OF as highly weighted features. For a reliable interpretation of the outcomes, there should be, for example, an association between the decrease in body weight and FGF21 concentration, and vice versa. This can be confirmed by the analysis of body composition in the groups that were switched to a LFD, and is in line with previous studies, although without using the correlation matrix [65]. Regarding the observed correlations, such as that between FGF21 concentration and mean velocity in the OF, these findings have no further consensus, as vertical activity and general mobility time were selected as the putative biomarkers for impaired obesity-related behavioural dysfunction. Thus, the correlation matrix was employed in this study to constrain predicted features and not vice versa. Therefore, we assume that the higher accuracy of the ML models based on the FS dataset corroborates the selected features, mainly FGF21 and body weight, as relevant biomarkers for impaired behaviour. In this context, the results underpin the hypothesis that FGF21 may serve as a putative biomarker for improved behaviour after weight reduction in obese mice.

However, the present study reveals limitations to some extent. The smell detection ability test (BPT) may have a high dropout rate and a high rate of false positive results [66]. Additionally, it was shown that obese-prone rats have an innate deficit with respect to sweet taste detection [67], which may explain the high dropout rates seen in the SPT. As the prediction of highly weighted features also included the latency time, this feature should be interpreted with caution. Another limitation of the study is the impact of the diet used in this study. The HFD was implemented to trigger inflammatory processes in the periphery [68], as well as in the brain [11]. However, a cafeteria diet more accurately mimics an obese human diet and the associated comorbidities, such as metabolic syndrome, but does not reveal inflammatory processes to the same extent as an HFD [69]. Nevertheless, this study provides potential insights for human studies. Based on the results, we speculate that instead of time-restricted eating, moderate exercise training and especially a change in eating behaviour could be of great interest in reducing body weight and fat. Additionally, further research of FGF21 in brain tissue and meta-analysis of both human and mice studies would provide essential insights into FGF21 interaction as a predictor or biomarker for impaired obesity-related behavioural dysfunction.

## 5. Conclusions

The variety of methodological approaches in this study leads to a compelling argument, with a recurring emphasis on the core groups and observations. To put the puzzle together, the evidence suggests that (i) the combination of LFD and TM improves body weight, circulating FGF21 concentration and behavioural parameters; (ii) the dietary switch to LFD and LFD with TM are very likely to interrupt the vicious circle of obesity; and (iii) FGF21 can be considered as a potential biomarker for improved behaviour after weight reduction, since an improvement in behaviour is associated with a lower FGF21 concentration. Moreover, collecting analogous, non-invasive parameters in humans would allow to verify whether FGF21 functions as a biomarker for improved locomotion and olfaction detection ability after weight reduction in obese mice.

## Figures and Tables

**Figure 1 nutrients-13-02916-f001:**
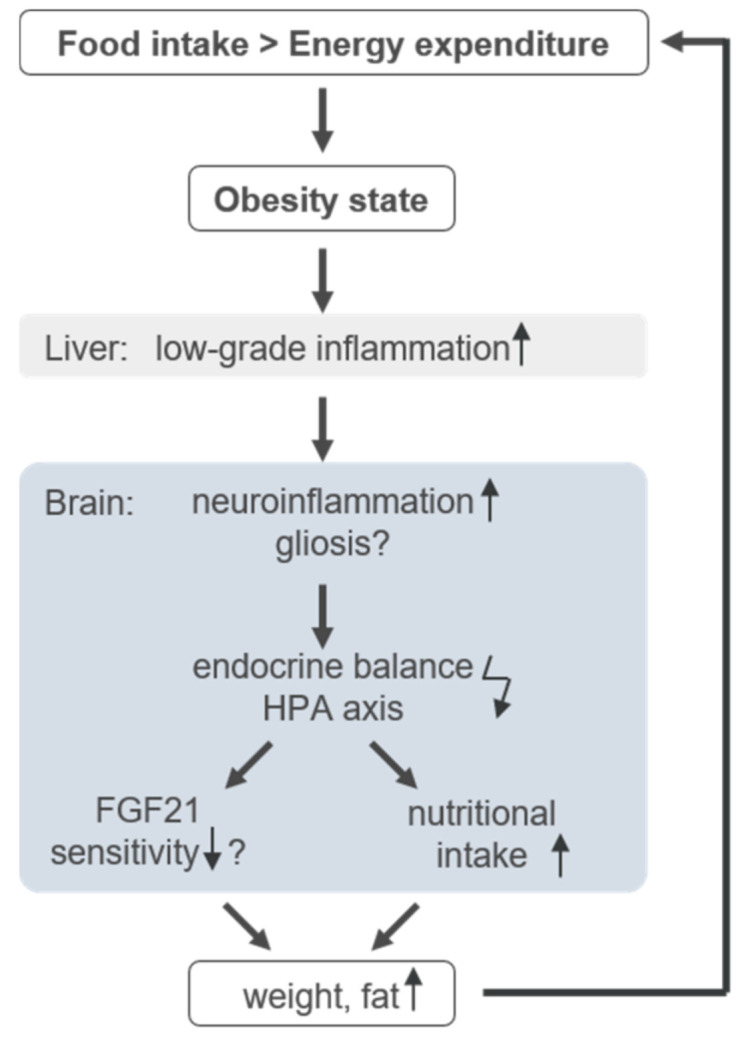
Infographic of the circulus vitiosus of obesity. Detailed information is provided in the text.

**Figure 2 nutrients-13-02916-f002:**
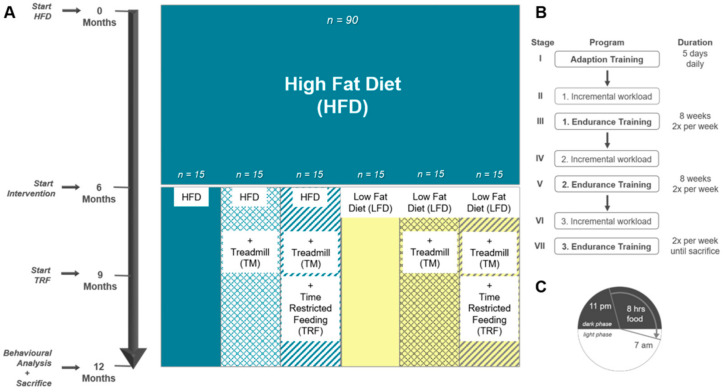
(**A**) Experimental design. A: Female C57BL/6J mice (*n* = 90) were fed 6 months on high-fat diet (HFD) to establish the model of DIO. Thereafter, mice were divided into 6 groups. The first group remained on HFD (*n* = 90). Groups two to six underwent an intervention. Second group: HFD plus treadmill exercise (TM; HFD/HFD + TM, *n* = 15); third group: HFD plus treadmill exercise, and time-restricted feeding (TRF; HFD/HFD + TM + TRF, *n* = 15); fourth group: diet change to a low-fat diet (LFD; HFD/LFD, *n* = 15); fifth group: diet change plus treadmill exercise (HFD/LFD + TM, *n* = 15), and sixth group: diet change, treadmill training and time-restricted feeding (HFD/LFD + TM + TRF, *n* = 15). When diet change was completed, treadmill training was applied. After 3 months of endurance exercise, time-restricted feeding was introduced. In the end, behaviour experiments were performed, and the mice were sacrificed. (**B**) Treadmill protocol consists of seven stages with three endurance sections. Prior to endurance training, an incremental workload test was performed in order to adjust the maximum velocity of the run. (**C**) Mice in the TRF group were restricted to food from 7 a.m. to 11 p.m. (16 h). Food supply was provided in the nocturnal active phase for 8 h.

**Figure 3 nutrients-13-02916-f003:**
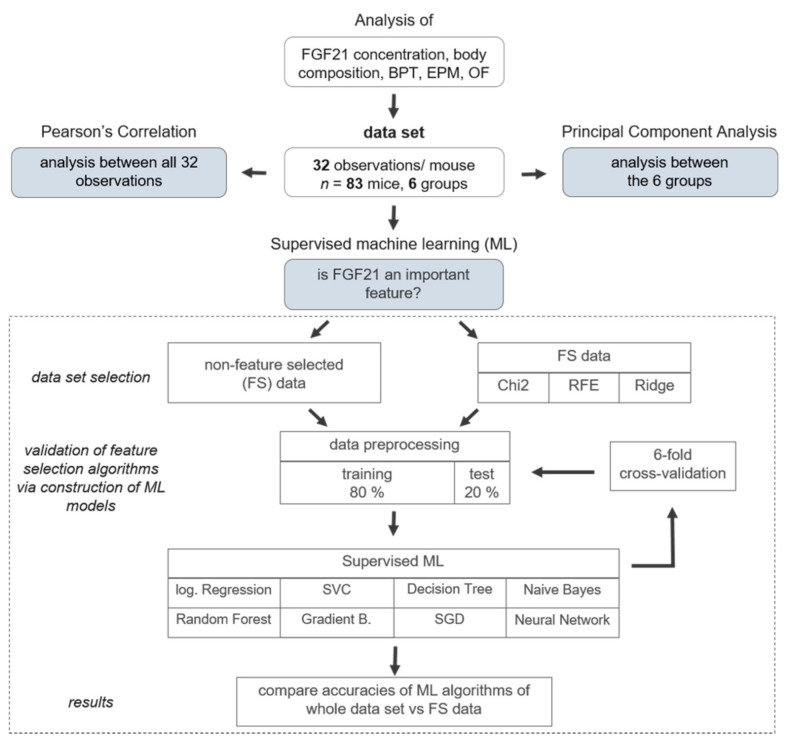
Analysis pipeline for dimensionality reduction and machine learning (ML) approaches. Analysis of Fibroblast Growth Factor (FGF) 21 concentration, body composition, buried pellet test (BPT), elevated plus maze (EPM), and open field (OF) yielded 32 observations with *n* = 83 mice divided into six groups. To determine pairwise correlations considering all 32 variables, Pearson’s Correlation was performed. To reveal new insights and relation between the six intervention groups, Principal Component Analysis (PCA) was conducted. Ultimately, to predict whether FGF21 is an important feature in the data set, three different FS algorithms were applied, namely Chi-Square (Chi2), Ridge Regularisation (RIDGE), and Recursive Feature Elimination (RFE). To assess the viability of the FS algorithms, eight different ML models were constructed-based either on FS common data or non-feature selected data. Therefore, the data set was split into a training (80%) and test (20%) data set. The following supervised ML algorithms were used: Logistic Regression (log. Regression), Support Vector Classifier (SVC), Decision Tree, Naive Bayes, Random Forest, Gradient Boosting (Gradient B.), Stochastic Gradient Descent (SGD), and Neural Network. Each model was additionally verified by 6-fold cross-validation and as a result, accuracies were compared between non-feature selected data set and FS data.

**Figure 4 nutrients-13-02916-f004:**
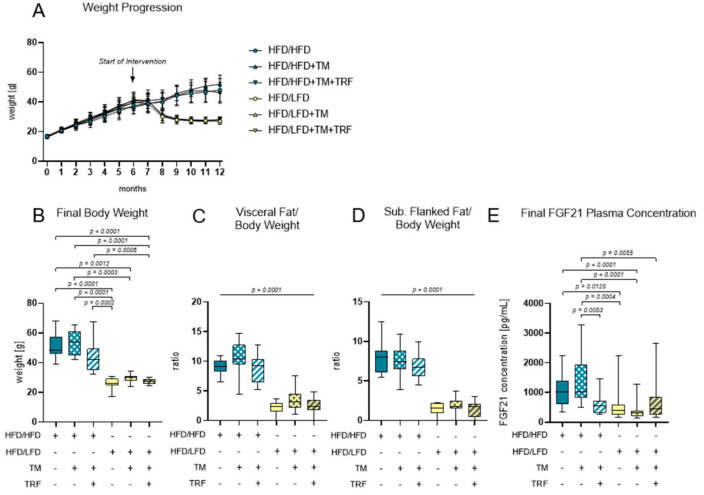
Body composition and FGF21 plasma concentrations. (**A**) Monthly weight progression with *n* = 90 mice at the beginning and *n* = 84 mice at final time point. (**B**) Final body weights [g] before euthanasia (HFD/HFD: *n* = 13, HFD/HFD + TM: *n* = 13, HFD/HFD + TM + TRF: *n* = 15, HFD/LFD: *n* = 13, HFD/LFD + TM: *n* = 15, HFD/LFD + TM + TRF: *n* = 13; total *n* = 82). (**C**) Ratio of visceral body fat deposits to body weight. (**D**) Ratio of subcutaneous flanked fat deposits to body weight. (**C**,**D**) All HFD/LFD groups showed a significant fat loss with *p* < 0.0001 when compared to all 3 HFD/HFD groups, respectively. (**E**) FGF21 plasma concentration [pg/mL] of final blood sample (HFD/HFD: *n* = 13, HFD/HFD + TM: *n* = 13, HFD/HFD + TM + TRF: *n* = 15, HFD/LFD: *n* = 13, HFD/LFD + TM: *n* = 15, HFD/LFD + TM + TRF: *n* = 14; total *n* = 83). Blue dots and box plots indicate HFD groups, yellow dots and box plots indicate diet change to LFD. The table below the figure displays the individual groups, respectively. Table is read from top to bottom, where ‘+’ denotes a diet or intervention, whereas ‘-’ does not refer to this parameter. Significance of differences between groups was tested with either Kruskal–Wallis test followed by Dunn’s post hoc test for multiple comparisons (B), Brown–Forsythe and Welch’s ANOVA with Tamhane T2 post hoc test for multiple comparisons (C: *F* value (*F*) = 51.52, Degree of Freedom (*DF*) = 5; D: *F* = 66.19; *DF* = 5) or by ordinary One-Way ANOVA with Tukey’s post hoc test for multiple comparisons (E: *F* = 9.765, *DF* = 5). Data are presented as mean ± SD and statistical significance was set at *p* < 0.05. Abbreviations: HFD: high-fat diet, LFD: low-fat diet, TM: treadmill, TRF: time-restricted feeding, FGF21: Fibroblast Growth Factor 21.

**Figure 5 nutrients-13-02916-f005:**
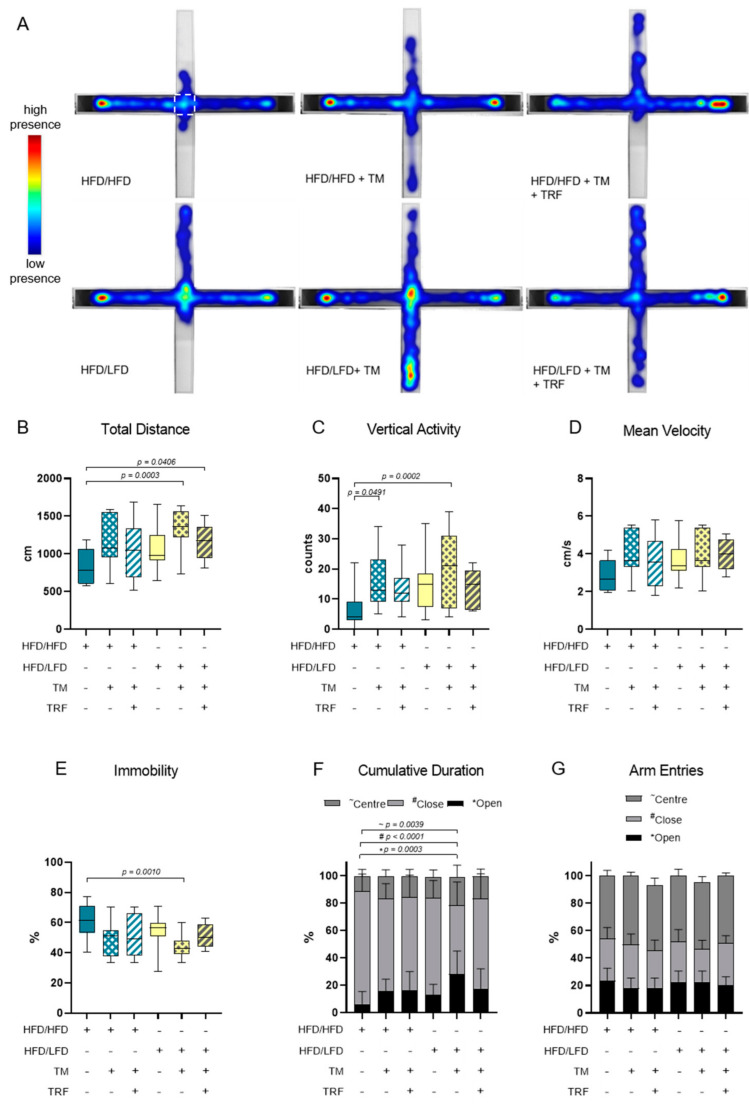
Elevated Plus Maze (EPM) analysis. The combination of a diet change and treadmill exercise leads to less anxiety-related behaviour. (**A**) Heatmap presentation of animal presence in the maze over 300 s whereas red colour reveals the highest residence and blue colour the lowest duration. The number of mice which exhibited the shown pattern amounted to: HFD/HFD: *n* = 7/13, HFD/HFD + TM: *n* = 5/13, HFD/HFD + TM + TRF: *n* = 7/15, HFD/LFD: *n* = 7/13, HFD/LFD + TM: *n* = 12/15, HFD/LFD + TM + TRF: *n* = 8/13, total *n* = 82. White dashed line indicates the centre. Boxplots indicate (**B**) total distance [cm] moved over 300 s, (**C**) vertical activity [counts] by manual counting of rearing, leanings and jumps, (**D**) mean velocity [cm/s] calculated from distance moved in maze over 300 s and (**E**) immobility pattern represented in [%]. (**F**,**G**) Column bars show cumulative duration [%] and arm entries [%] in centre (dark grey ~), in closed arm (light grey #) or in open arm (black *), respectively. Significance of differences were tested by Kruskal–Wallis with Dunn’s post hoc test for multiple comparisons (F, G~) or by ordinary one-way ANOVA with Tukey’s post hoc test for multiple comparisons (B: F = 2.043, DF = 5; C: F = 4.504, DF = 5; D: F = 2.149, DF = 5, E: F = 3931, DF = 5; G*: F = 2.043, DF = 5; G#: F = 2.006, DF = 5). Data are presented as mean ± SD and statistical significance was set at *p* < 0.05. Abbreviations: HFD: high-fat diet, LFD: low-fat diet, TM: treadmill, TRF: time-restricted feeding.

**Figure 6 nutrients-13-02916-f006:**
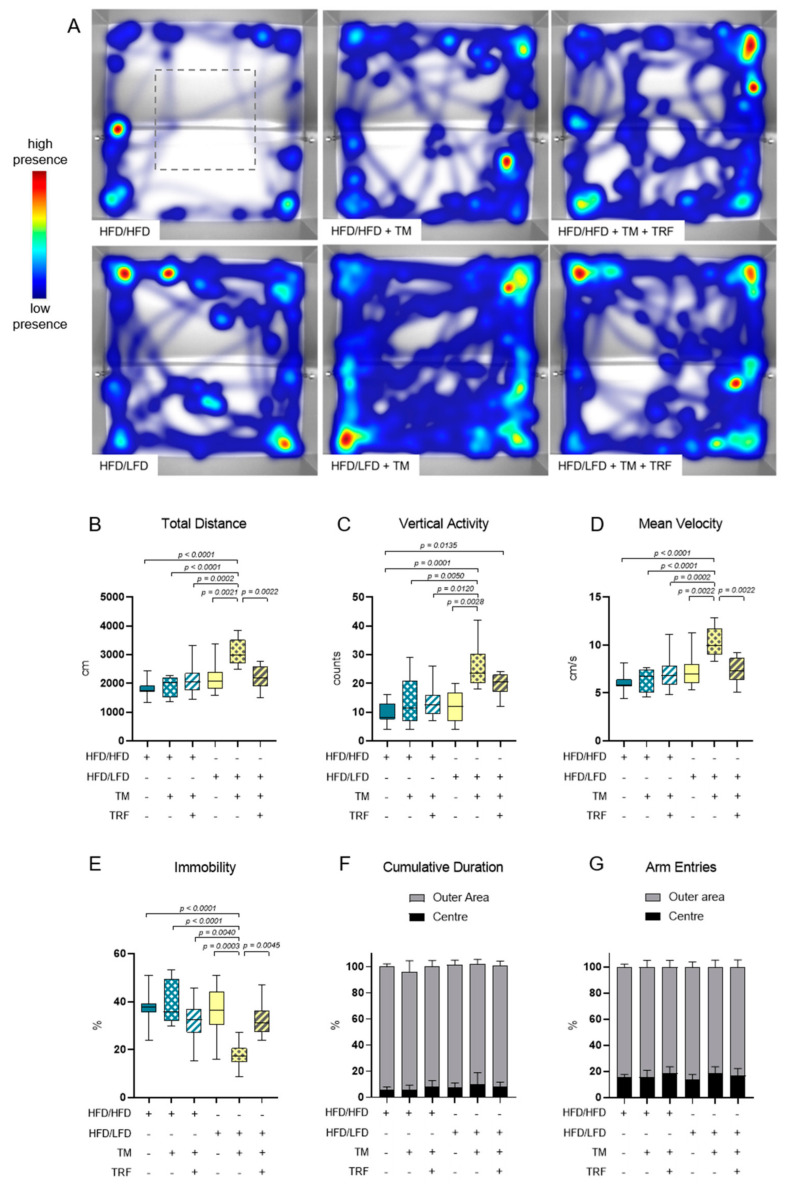
Open Field (OF) analysis. The combination of a diet change and treadmill exercise leads to more locomotor activity. (**A**) Heatmap presentation of animal presence in maze over 300 s whereas red colour reveals highest residence and blue colour lowest duration. The number of mice that exhibited the shown pattern amounted to: HFD/HFD: *n* = 5/9, HFD/HFD + TM: *n* = 3/8, HFD/HFD + TM + TRF: *n* = 5/10, HFD/LFD: *n* = 3/8, HFD/LFD + TM: *n* = 5/10, HFD/LFD + TM + TRF: *n* = 4/9; total *n* = 54. Grey dashed line indicates the centre. Boxplots indicate (**B**) total distance [cm] moved over 300 s, (**C**) vertical activity [counts] by manual counting of rearing, leanings and jumps, (**D**) mean velocity [cm/s] calculated from distance moved in maze over 300 s and (**E**) immobility pattern represented in [%]. (**F**,**G**) Column bars show cumulative duration [%] and arm entries [%] in the outer area (light grey) and the centre (black), respectively. Significance of differences were tested by Kruskal–Wallis with Dunn’s post hoc test for multiple comparisons (F) or by ordinary one-way ANOVA with Tukey post hoc test for multiple comparisons (B: F = 9.756, DF = 5; C: F = 7056, DF = 5; D: F = 9.710, DF = 5; E: F = 8.886, DF = 5; G: F = 1.420, DF = 5). Data are presented as mean ± SD and statistical significance was set at *p* < 0.05. Abbreviations: HFD: high-fat diet, LFD: low-fat diet, TM: treadmill, TRF: time-restricted feeding.

**Figure 7 nutrients-13-02916-f007:**
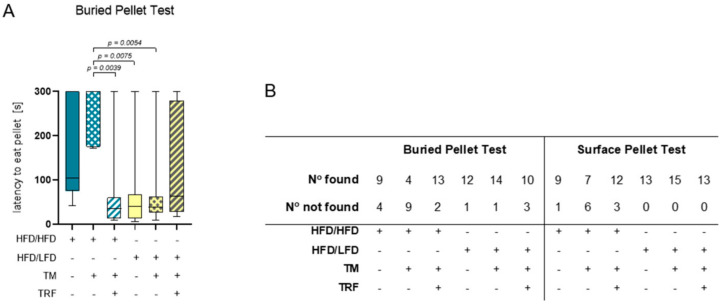
Results of the buried pellet test and surface pellet test. (**A**) Impaired sense of smell altered depending on intervention parameters. Boxplots indicate performance of HFD/HFD (*n* = 13), HFD/HFD + TM (*n* = 13), HFD/HFD + TM + TRF (*n* = 15), HFD/LFD (*n* = 13), HFD/LFD + TM (*n* = 15) and HFD/LFD + TM + TRF (*n* = 13). Significance of differences was tested by Kruskal–Wallis with Dunn’s post hoc test for multiple comparisons. Data are presented as mean ± SD and statistical significance was set at *p* < 0.05. (**B**) Results of finding and not finding pellets in the buried pellet test and surface pellet test. Abbreviations: HFD: high-fat diet, LFD: low-fat diet, TM: treadmill, TRF: time-restricted feeding.

**Figure 8 nutrients-13-02916-f008:**
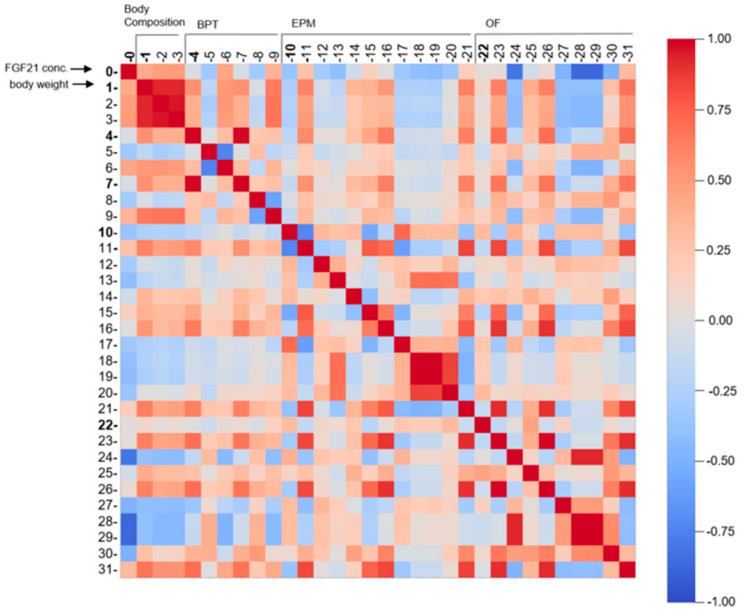
Pearson’s correlation represented as a multidimensional heatmap. Pearson’s correlation plot visualises the correlation values between all 32 acquired parameters. Scale bar represents the range of the correlation coefficients displayed. Red colour (1.00–0.70) indicates a strong positive correlation, blue colour (−0.70–−1.00) reveals a strong negative correlation, whereas light colour above 0.40 or under −0.40 indicates moderate correlation. Grey colour with coefficient approximately 0 displays no correlation. Numbers correspond to experiments as followed: 0 = FGF21 concentration (*n* = 82); 1–3 = Values of body composition (*n* = 80–81); 4–9 = Observations of BPT (*n* = 81); 10–21 = Observations of EPM (*n* = 76); 22–31 = Observations of OF (*n* = 53; see Appendix A for more information). All exact correlation values are displayed in Appendix A. Abbreviations: FGF21: Fibroblast Growth factor 21, BPT: buried pellet test, EPM: elevated plus maze, OF: open field.

**Figure 9 nutrients-13-02916-f009:**
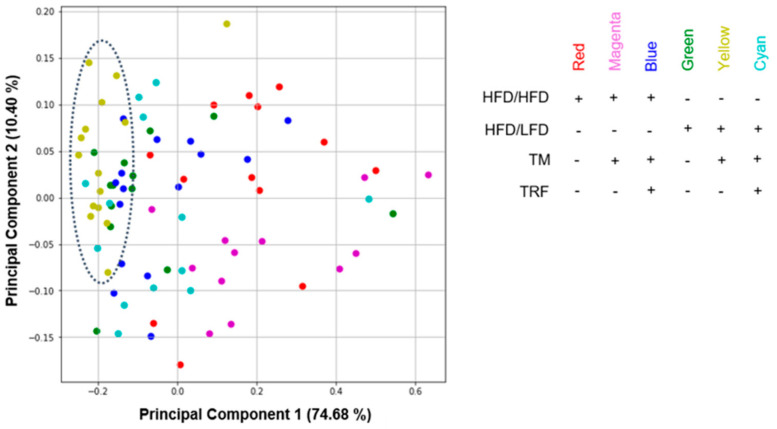
Dot plot of principal component analysis (PCA). All parameters from previously acquired experiments were further used for PCA construction (*n* = 83 mice with 32 observations, see Appendix A for more information). Components from HFD/LFD + TM (yellow) and in parts from HFD/LFD (green) concentrate more on the left part of the diagram revealing less variance and more similarity (dotted circle). All PCA variables are listed in Appendix A. Abbreviations: HFD: high-fat diet, LFD: low-fat diet, TM: treadmill, TRF: time-restricted feeding.

**Figure 10 nutrients-13-02916-f010:**
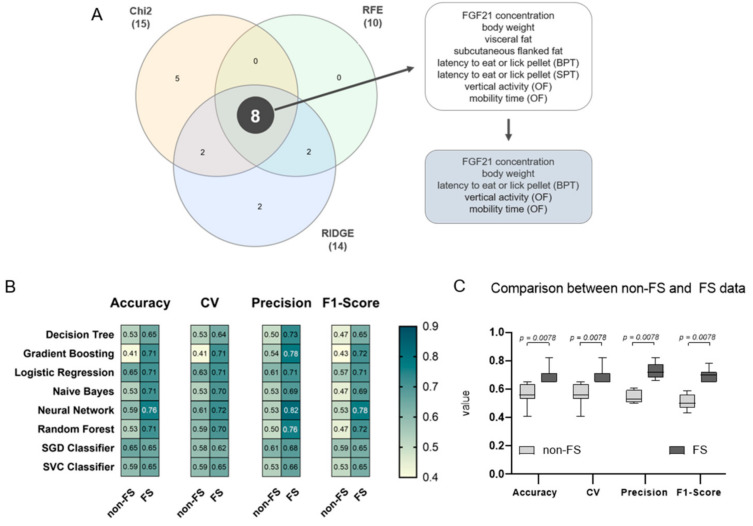
Feature extraction and validation of feature selected (FS) data. (**A**) To extract weighted features of the data set, three different FS algorithms were applied (Chi-Square (Chi2), Recursive feature elimination (RFE), and Ridge regularisation (RIDGE)). Generated features are represented in a Venn diagram revealing eight common features, namely Fibroblast Growth Factor (FGF) 21 concentration, body weight, subcutaneous flanked fat, visceral fat, latency to eat or lick the pellet in the BPT and SPT, vertical activity, and mobility time in the OF. These features can be further restricted in a biological context yielding five features, which are FGF21 concentration, body weight, and latency to eat or lick the pellet in the BPT, vertical activity, and mobility time in the OF. (**B**) Heatmap representation of the classification report for each ML model, namely Decision Tree, Gradient Boosting, Logistic Regression, Naive Bayes, Neural Network, Random Forest, SGD Classifier, and SVC Classifier. Dark petrol blue colour indicates a higher accuracy, mean accuracy in 6-fold cross-validation (CV), precision or F1-score; yellow colour indicates the contrary. For a complete classification report with the best parameters see Appendix A. (**C**) Models based on the FS data set (dark grey) scored significantly higher compared to models based on the non-FS data set (light grey) with *p* = 0.0078, respectively. Significance of differences was tested by Wilcoxon Signed Rank Test where theoretical medians (*tm*) were set to the mean of the corresponding non-FS data: *tm* of accuracy = 0.56; *tm* of CV = 0.56; *tm* of precision = 0.54; and *tm* of F1-score = 0.51. Data are presented as mean ± SD and statistical significance was set at *p* < 0.05.

**Table 1 nutrients-13-02916-t001:** List of applied algorithms with their respective implementations in Python.

Task		Library	Class
handling missing values	stratified mean	pandas, numpy	
correlations between all 32 observations	data preprocessing	sklearn.preprocessing	Normalizer
	Pearson’s Correlation	scipy.stats.pearsonr	
	visualization	matplotlib.pyplot	
relations between the 6 intervention groups	data preprocessing	sklearn.preprocessing	Normalizer
	PCA	sklearn.decomposition	PCA
	visualization	matplotlib.pyplot	
feature selection	Chi2, data preprocessing	sklearn.preprocessing	MinMaxScaler
	Chi2	sklearn.feature_selection	chi2, SelectKBest
	RFE, data preprocessing	sklearn.data sets	make_friedman1
	RFE	sklearn.feature_selection sklearn.svm	RFE SVR
	RIDGE, data preprocessing	sklearn.linear_model	Ridge
	RIDGE	sklearn.feature_selection	SelectFromModel
machine learning algorithms	data preprocessing	sklearn.preprocessing	Normalizer, StandardScaler
	logistic Regression	sklearn.linear_model	LogisticRegression
	SVC Classifier	sklearn.svm	SVC
	Decision Tree	sklearn.tree	DecisionTreeClassifier
	Naive Bayes	sklearn.naive_bayes	GaussianNB
	Random Forest	sklearn.ensemble	RandomForestClassifier
	Gradient Boosting	sklearn.ensemble	GradientBoostingClassifier
	SGD Classifier	sklearn.linear_model	SGDClassifier
	Neural Network	sklearn.neural_network	MLPClassifier
	cross validation	sklearn.model_selection	StratifiedKFold
	hyperparameter tuning	sklearn.model_selection	GridSearchCV

## Data Availability

Full data table and all coding sections were uploaded on 21 April 2021 and modified on 22 August 2021. https://github.com/IEC-2020/Intervention (accessed on 21 April 2021).

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
