# Peer review of "Fibroblast Growth Factor 21 as a Potential Biomarker for Improved Locomotion and Olfaction Detection Ability after Weight Reduction in Obese Mice"

_nutrients, 2021, doi:10.3390/nu13092916_

Round 1

Reviewer 1 Report

  1. The authors state in abstract that “In this study, we showed that the combination of dietary change with treadmill exercise resulted in body weight reduction…” As indicated in Figure 3B, body weight was reduced in all LFD groups to a comparable level and was not affected by TM or TM+TRF. Similarly, TM and/or TRF did not lead to a further reduction of FGF21 in all LFD groups (Figure 3E). Thus, it appears that body weight and FGF21 were reduced by LFD alone, but not combination.
  2. The authors conclude that FGF21 (by using some methodological approaches) can be a putative biomarker after tackling obesity in mice. However, FGF21 was not indicative of body weight changes (e.g., TM+TRF in HFD, body weight remained unchanged, but FGF21 reduced) nor represented behavior improvement (e.g., LFD alone and LFD+TM+TRF, FGF21 levels were decreased, but no improvement in behavioral tests). The authors should elaborate more on this discrepancy between the analytical prediction and the results from their mouse models.
  3. The manuscript would benefit from providing data regarding the effects of dietary treatments on the expression of FGF21 in the brain and/or the pathological changes of the brain, which may support the findings of the behavioral improvement.
  4. Given that FGF21 has been reported to be a potential biomarker for many other diseases and health conditions, the authors should discuss the specificity of FGF21 as a putative biomarker in their mouse models.
  5. Since the control mice at a regular rodent chow were missing in the current study, the normal behavioral baseline is not available. Thus, the author should elaborate more about the “improved behavior” in mice with HFD/LFD+TM.

Reviewer 2 Report

Thank you for submitting your manuscript for peer review. It was indeed a pleasure to review your submission. The subject of manuscript is certainly interesting and of interest of the readership; and the work has been conducted with care and enthusiasm, benefiting from multidisciplinary and innovative design.

While Methods, results and discussions are often clear and well presented; the abstract and introduction require revision and improvement to be comparable with the rest of the manuscript. The following amendments are therefore recommended before further consideration of the manuscript for publication:

  1. The title needs amendment and clarification. What behaviour? in what way and direction?
  2. The abstract requires major amendment, giving further focus on answering to three key questions of what was done, found and concluded? The key findings needs to be outlined in the abstract. I appreciate that there are too many points to report, but the most important findings in relation to the aim should be selected and presented.
  3. Line 31-33, the statement needs clarification and rewording.
  4. Lines 59-69, I strongly recommend production of a figure, schematic or infographic to summarise the mechanism interlinking inflammation, obesity and FGF21. 
  5. Lines 78-82, the aim and hypotheses may need revision (consider appropriateness of words such as endure, counteracting, etc).
  6. Lines 83-96 come out of context. The reader would legitimately ask what prediction and what data/datasets. Further information and clarification is required to set the scene and introduce the approach of this study.
  7. The introduction does not review the body of knowledge. Review of literature and more specifically review of the most relevant studies that contribute to the conceptual framework of this research is required. I suggest adding an extensive table with previous research, findings, strengths and limitations to produce a summary of the previous literature, while the current study should be justified as the logical consequence of the gap in the body of literature discussed within the table.
  8. Methods are generally well written; although the section can be improved via adding some key information about the research design, ethical approval, inclusion and exclusion criteria and the explanation of the sample size being reduced to 83. The well written section also be further improved by addition of some figure, photographs to further clarify the methods.
  9. In discussions, the limitations of the study need to be clearly articulated. 
  10. At the end of the day, the findings would become far more meaningful, when interlinked with human studies. So, the authors might want to add a paragraph in discussions, to compare these findings with limited findings of the human studies, and explain the potential learnings for human studies.
  11. In conclusions, you are legitimately trying to summarise the findings and put the puzzle together. The article may benefit from a final table/figure summarising step by step (as per excellent results presented) what do we know, and what this study add.

Reviewer 3 Report

This manuscript describes the effects of three different interventions alone, or in combination, on body weight, adiposity, serum FGF21 concentrations, and behavior in diet-induced obese mice. The interventions: low-fat (and presumably lower energy) diet, treadmill exercise, and time-restricted feeding were investigated. The experimental procedures were thoroughly described. Only dietary intervention reduced body weight. Similarly, the dietary intervention reduced serum FGF21 concentrations compared to the HFD control group. Treadmill exercise or time-restricted feeding provided no significant reductions in FGF21 compared to the HFD control group or the LFD control group. Diet alone did not affect behavioral tasks compared to the HFD control group. However, LFD in combination with treadmill exercise or time-restricted feeding improved performance on some behavior tasks. These data are interesting, especially considering the effects of diet, obesity, and these interventions on cognition and behavior are unclear. However, a major focus of this manuscript is the authors’ claim about the potential for FGF21 as a “biomarker”. Yet it is not clear what they are claiming FGF21 is a biomarker for. This needs to be clarified. Please see additional points below.

There was a failure to cite any work by the lab pioneering much of the work in FGF21 in energy metabolism. The Kliewer/Mangelsdorf lab was not cited. Much of the background provided about FGF21 in the introduction section should have referenced their work. For example, PMID# 25130400 should be cited in line 55 with the other citations.

Line 30 and Line 530: Exercise is not actually a successful intervention for obesity.  I also think TRF is not an accepted method for weight loss. One of the most successful interventions for obesity is left off this list: bariatric surgery.

Line 34: “last” should be “lasts”

Title, line 34, throughout “tackling obesity” is a vague phrase. If the authors are referring to weight loss, that should be stated clearly.

Line 39 “Further analysis of the data by principal component analysis revealed the importance of the combination of a dietary change and treadmill exercise.” What are the authors referring to here? What importance?

Line 34: “…whether FGF21 lasts as a biomarker after tackling obesity”.  Can the authors clarify what FGF21 is a biomarker for? Same comment for lines 43, 81, 94, 302, 484, 508, 534, 537, 615, 627, 628 and 637. 

Line 65: It is my understanding that reference #13 makes no claim about inflammation reducing FGF21 sensitivity.

Line 72 implies that obesity causes these cognitional behavior changes. However, reference #18 points out that it is diet, independent of body weight that causes these changes.

Line 237 and 238: What do the authors mean by “weight control” and “Body weight was controlled weekly”? If weight was simply measured, just say “measured”.

Line239: At what time were the mice euthanized? There were a lot mice that needed to be euthanized; were different groups euthanized at different times or where the mice randomized? This is important as there were groups of mice that were experiencing time-restricted feeding. It is known that FGF21 concentrations change quickly in response to fasting and refeeding, so timing of sacrifice is important to know.

Figure 3: Please include a picture alongside Figure 3A indicating which color/symbol corresponds to which intervention group.

Please remove references to “blue” or “yellow” from the text of the manuscript. It’s fine in the figure legends.

Line 340: “Noteworthy, the HFD/HFD+TM+TRF group also displayed a significant reduction in FGF21 concentration…”. Can the authors please clarify that this was not different from HFD/HFD. This was only a reduction compared to HFD/HFD+TM.

Line 556: “TRF was shown to reverse the consequences of obesity.”. What is this sentence referring to specifically?

Line 633: “circular” should be “circulating”.

Line 623: “…have been selected as the more putative biomarker”… Biomarker for what?
